# Biointeraction of Erythrocyte Ghost Membranes with Gold Nanoparticles Fluorescents

**DOI:** 10.3390/ma14216390

**Published:** 2021-10-25

**Authors:** Víctor Gómez Flores, Alejandro Martínez-Martínez, Jorge A Roacho Pérez, Jazzely Acosta Bezada, Francisco S. Aguirre-Tostado, Perla Elvia García Casillas

**Affiliations:** 1Instituto de Ingeniería y Tecnología, Universidad Autónoma de Ciudad Juárez, Ciudad Juárez 32310, Mexico; victor.gomez@uacj.mx (V.G.F.); jazz_bezada@hotmail.com (J.A.B.); 2Instituto de Ciencias Biomédicas, Universidad Autónoma de Ciudad Juárez, Ciudad Juárez 32310, Mexico; alejandro.martinez@uacj.mx; 3Departamento de Bioquímica y Medicina Molecular, Facultad de Medicina, Universidad Autónoma de Nuevo León, Monterrey 64460, Mexico; alberto.roachoprz@uanl.edu.mx; 4Centro de Investigación en Materiales Avanzados, S. C., Unidad Monterrey, Apodaca Nuevo León, Monterrey 66628, Mexico; servando.aguirre@cimav.edu.mx

**Keywords:** nanomedicine, AuNPs, membrane, erythrocyte ghosts, fluorescence, hemoglobin

## Abstract

The application of new technologies for treatments against different diseases is increasingly innovative and effective. In the case of nanomedicine, the combination of nanoparticles with biological membranes consists of a “camouflage” technique, which improves biological interaction and minimizes the secondary effects caused by these remedies. In this work, gold nanoparticles synthesized by chemical reduction (Turkevich ≈13 nm) were conjugated with fluorescein isothiocyanate to amplify their optical properties. Fluorescent nanoparticles were deposited onto the surface of hemoglobin-free erythrocytes. Ghost erythrocytes were obtained from red blood cells by density gradient separation in a hypotonic medium and characterized with fluorescence, optical, and electron microscopy; the average size of erythrocyte ghosts was 9 µm. Results show that the functional groups of sodium citrate (COO-) and fluorophore (-N=C=S) adhere by electrostatic attraction to the surface of the hemoglobin-free erythrocyte membrane, forming the membrane–particle–fluorophore. These interactions can contribute to imaging applications, by increasing the sensitivity of measurement caused by surface plasmon resonance and fluorescence, in the context of biological membranes.

## 1. Introduction

Recently, sensitive and fast techniques have been developed in biomedical applications, such as chemiluminescence, enzymatic techniques, chemical sensing, and spectrofluorimetry, which enable the sensing and detection of different organic structures. There are several materials used in nanosensors such as carbon nanotubes and metal-organic frameworks, [1,2]. Gold nanoparticles (AuNPs) offer an alternative for the development of new technologies in different fields, particularly in the context of medicine (nanomedicine). AuNPs have sparked interest due to their capacity to conjugate with organic structures and interact with biological systems [3]. Some possible applications of these bioconjugates include the prevention of atherosclerotic plaque formation [4], controlled drug release [5], photodynamic therapy [6,7,8,9,10,11] in cancer treatments [12,13,14], antibacterial agent [15], induction of specific immune responses [16], and treatment of diabetes by insulin delivery [17] that involves the coupling of oligonucleotides in gene regulation [18]. These applications depend on the size and shape of the particles [19], and their properties allow its use as a sensor. Under light irradiation in the visible spectrum, the free electrons in the metal immediately form an electromagnetic field and begin a collective oscillation relative to the network of positive ions at a frequency that coincides with the incident light, which is known as a surface plasmon resonance (SPR) [20]. The addition of fluorophores to AuNPs enhances their spectral properties, allowing use in medical diagnostics and biotechnology. Fluorophores interact with free electrons of metal nanoparticles increasing the resonance energy, which due to the strong emission allows the quantification of molecules or other ligands [21].

The integration of fluorescent nanoparticles into biological structures with the capacity to host and release a drug improves their performance in biomedical applications [21,22]. Red blood cells, also known as erythrocytes, have been used as biological carriers due their biophysical properties [23,24,25,26]; these are enucleated cells without organelles with a diameter of 8–9 µm; they are flexible due to the properties of the plasma membrane and can circulate freely in the human body. A variant of these red blood cells is the erythrocyte ghost [27,28,29,30], which is a membranous structure free of hemoglobin that originates when the erythrocyte is exposed to a hypotonic medium, causing cell cytolysis and releasing hemoglobin into the medium (hemolysis), giving it the capacity to encapsulate a drug. These phantom erythrocyte membranes serve as a “camouflage” system [30,31] because they do not generate an immune response, are highly compatible, and keep the drug isolated from the environment until they are placed within a target organ. The structure of these membranes from red cells has been studied as a membrane model due to the simplicity of the membrane and its easy purification in healthy humans [32].

In this work, we attached fluorescent AuNPs onto the membranes of ghost erythrocytes in order to use them as drug carriers capable of detecting and sensing. A therapeutic agent can be loaded on the membrane; in addition to the optical properties of AuNPs, it works as a therapeutic agent. These autologous constructs can be used in Parkinson’s, which is a disease where significant erythrocyte extravasation has been diagnosed in the course of this illness, especially in the postcommissural putamen; thus, these affected regions have been proposed as a possible therapeutic entrance for the pharmacological treatment of Parkinson´s disease [33]. Here, we propose that erythrocyte ghosts can act as carriers for drug treatment of any neurological disease such as Parkinson’s that involves blood-brain barrier fracture—for instance, in the treatment of accidents such as poisoning, where it is known that erythrocytes are extravasated in the area postrema [34] and in cerebrovascular accidents that involve brain hemorrhages. This construct has the capacity to interact with human tissue, showing great adaptability and affinity, reducing most or all the toxicity or damage that may result from the juxtaposition of the external material with the biological entity.

## 2. Materials and Methods

### 2.1. Preparation of Gold Nanoparticles

The synthesis of AuNPs was carried out by chemical reduction, following the procedure described in the Turkevich method [35,36]. First, 18 mL of chloroauric acid (HAuCl_4_·3H_2_O 0.5 mM, Sigma Aldrich^®^, St. Louis, MO, USA) were heated to 75 °C while stirring constantly, and subsequently, 2 mL of sodium citrate at 38.8 mM (Na_3_C_6_H_5_O_7_·2H_2_O JT Baker^®^, Allentown, PA, USA) was added. The solution was heated to boiling point. The solution color changed from a crystalline tone to ruby red, indicating the formation of nanoparticles, as shown in Equation (1). After preparation, the nanoparticles were washed by centrifugation at 12,000 rpm × 10 min; then, the supernatant was removed, and 10 mL of distilled water was added to the pellet in order to prepare samples ranging between 150 nM and 10 nM, which were subsequently resuspended. The solution was allowed to cool to room temperature and stored for later use.
4HAuCl_4_(aq) + 3C_6_H_5_O_7_^3−^(aq) + 3H_2_O(l)→4Au(s) + 6CO_2_(g) + 3C_4_H_6_O_4_(aq) + 7H^+^(aq) + 16Cl^−^(aq)(1)

#### Preparation of Fluorescein Isothiocyanate-Coated Gold Nanoparticles (AuNPs-FI)

AuNPs were coated using the Elfeky method [37]. The scheme for the preparation of AuNPs is shown in Figure 1. Two precursor solutions were used: 18 mL of chloroauric acid at 0.5 mM (HAuCl_4_·3H_2_O, Sigma Aldrich^®^, St. Louis, MO, USA) and 2 mL of sodium citrate at 38.8 mM (Na_3_C_6_H_5_O_7_·2H_2_O, JT Baker^®^, Allentown, PA, USA) mixed with 0.006 g (1.18 × 10^−5^ M) of fluorescein isothiocyanate (FI) (C_21_H_11_NO_5_S Thermo Scientific™, Waltham, MA, USA). First, the chloroauric acid solution was brought to boiling point and agitated by magnetic stirring. Later, sodium citrate-FI solution was added and heated to 75 °C for 30 min. The color changed from crystalline to a red–green color, indicating the formation of AuNPs-FI. Subsequently, 10 washes were carried out at 9000 rpm × 30 min in order to remove traces of FI.

### 2.2. Nanoparticle Characterization

Particle morphology was determined by scanning electron microscopy (JSM-7000F, JEOL, Akrishima, Japan) and transmission electron microscopy (Hitachi 7700, TEM, Tokyo, Japan). For SEM analysis, the various samples were spread on a carbon tape slide. Once the sample was dry, a secondary electron image and energy-dispersive X-ray spectroscopy (X-act EDS detector from Oxford, Instruments, Abingdon, UK) study were performed. Particle size distribution and zeta potential were measured with dynamic light scattering equipment (DLS) (Nanotrack Wave II, Microtrac, Haan, Germany). UV-Vis absorbance was measured by nanodrop spectroscopy (Thermo Fisher Scientific, Waltham, MA, USA) at 450 nm. The crystal structure was made by means of X-ray diffraction (XRD PANalytical, X’Pert PRO, EA, Almelo, The Netherlands) Anode material: Cu, K-Alpha1 [Å] = 1.54060, Generator Settings: 25 mA, 35 kV). Hemolysis was determined according to ASTM F756-17 (Standard Practice for Assessment of Hemolytic Properties of Materials) [38].

### 2.3. Isolation of Erythrocyte Ghost Membranes

Hemoglobin-free plasma membrane was obtained from erythrocytes extracted from the blood of healthy donors by venipuncture in ethylenediaminetetraacetic acid (EDTA) anticoagulant tubes (Vacutainer^TM^, Becton, Dickinson and Company, Franklin Lakes, NJ, USA). We followed the protocol described by Grigoruţa [39] to separate red cells. Whole blood was diluted with phosphate-buffered saline solution (PBS) and EDTA 1:1 (137 nM NaCl, 8.2 mM Na_2_HPO_4_, 1.5 mM KH_2_PO_4_, 3.2 mM KCl, and 4 mM EDTA, pH 7.4). This dissolution was slowly placed onto a bed of 40% PBS-EDTA, 57.3% Percoll^TM^ (Cytivia, St. Louis, MO, USA) 2.7% 10 X PBS (to reach a final ratio of 9:1). After centrifugation at 1500 rpm for 30 min, erythrocyte cells were washed three times with 1X PBS solution. In order to obtain erythrocyte ghosts, we followed the methodology modified by Niggli et al. [40]. Red cells were washed twice with 5 volumes of 130 mM KC1, 20 mM Tris-C1, pH 7.4. The cells were hemolyzed using 5 volumes of 1 mM K-EDTA, 10 mM Tris-C1, pH 7.4, and centrifuged at 18,000 rpm for 10 min. The ghosts were washed four times with PBS buffer without EDTA. The hemoglobin-free ghosts were stored in 1.5 mL microtubes at 4 °C.

### 2.4. Bio-Interaction

The interaction of AuNPs with and without FI with erythrocyte ghosts was carried out according to the methodology described by Che et al. [35]. A solution containing 10 µL of AuNPs (4.08 nM) and AuNPs-FI (0.26 nM) was dispersed using ultrasound at 0.5 cycles, 80% amplitude (Heilsher, Ultrasound Technology, Teltow, Germany), and then incubated with 50 µL of hemoglobin-free membranes (1:5 ratio) for 24 h at 37 °C under constant agitation. 

## 3. Results

### 3.1. AuNPs

AuNPs synthesized by chemical reduction have an average diameter of 13 nm ± 2 nm, which was confirmed using the DLS technique. Figure 2 shows a normal and narrow particle size distribution graph. These particles have a face-centered cubic crystalline structure according to JCPDS No. 03-035-1870 [41], and the diffracted crystallographic planes 111, 200, 220, and 311 were identified using Bragg´s law as shown in Figure 3. Scherrer’s law showed the crystal size to be 17.94 nm. The absorption peak for the surface plasmon at 525 nm is characteristic for AuNPs of this particle size (Figure 4). 

Au nanoparticle geometry is spherical, as apparent in the SEM and TEM micrographs in Figure 5a–c; this spherical shape is characteristic of the Turkevich method [36]; EDS spectroscopy (Figure 5d) shows an intense peak due to Au, and other elements are present as the result of residues left in the sample, following the synthesis process of the nanoparticles.

The particle size obtained by SEM (13.47 ± 0.4 nm) is shown in Figure 5e, and DLS (13.07 ± 1.3 nm) is similar to the crystal size obtained by DRX (17.94 nm), so evidently, the nanoparticles are monocrystalline. The zeta potential of the Au nanoparticle was −15.3 ± 1.3 mV at pH = 7. 

### 3.2. AuNPs-FI

FI was adhered to AuNPs, which is a fluorescent dye with λ = 490 nm emission and a maximum peak at λ = 550 nm. In Figure 6, we present the UV-Vis absorption graph of AuNPs-FI. Here, the FI band has adhered to the gold nanoparticle band, so the absorption peak of AuNPs shifts to λ = 527 nm due to the Resonance Fröest, demonstrating the union between the metal and the organic molecule (AuNPs-FI). Figure 7 shows a bimodal particle size distribution of AuNPS-FI. The main population consists of dispersed nanoparticles with an average particle size of 9 nm, the second population between 1000 and 10,000 nm includes the agglomerates that are formed in the synthesis when adding FI. The hydrodynamic size obtained by DLS was 30 ± 2 nm.

The SEM and TEM images in Figure 8 show slightly large particles when the fluorophore is added (10–27 nm) due to their organic nature, which is a feature that is not possible to discern in microscopy analysis. The increase in hydrodynamic size indicates a coupling between FI and AuNPs. FI changed the zeta potential to −25.7 ± 1.3 mV at pH = 7.

In visible light, AuNPs are a red ruby solution (Figure 9a) that turns dark when fluorescein isothiocyanate binds (Figure 9b); however, under UV light, they show their fluorescence (Figure 9c), which facilitates the detection of a bioanalyte. 

#### Hemolytic Assays

Figure 10 shows the hemolysis graph for different AuNP concentrations. The greatest AuNP concentration (10,000 µg/mL) has a hemolytic degree close to 2%. This decreases proportionally in accordance with AuNPs concentration, so that in terms of the ASTM F756-17 standard [38], this is considered to be a non-hemolytic material. 

### 3.3. Erythrocyte Ghost 

A blood smear was taken in order to observe the red blood cells under an optical microscope (Figure 11a); the erythrocytes show a biconcave shape. The hemoglobin-free erythrocyte membranes (Figure 11b) were obtained from peripheral whole blood. The process of elimination of hemoglobin protein (Figure 11c,d) from erythrocytes was verified by UV-Vis after each wash with lysis buffer, up until the addition of the final 1X PBS solution. Hemoglobin has three characteristic absorption peaks at λ = 575, 540, and 412 nm, respectively. The measurements were as follows: (α) λ = 578 nm, (β) λ = 543 nm, and (γ) λ = 415 nm, as presented in Figure 12a, confirming the presence of the characteristic hemoglobin bands. During processing, washing eliminates hemoglobin (Figure 12b), so that absorption decreases after each wash with lysis buffer from 100% to 8.71%, corroborating the loss of hemoglobin. A slight increase is noted after the penultimate wash, but this is due to the recurrences that continue prior to the addition of PBS solution, as shown in Figure 12c. However, it is very difficult to eliminate hemoglobin from human red blood cells, so small remnants remain. The reddish solution that became clear during the process (Figure 11c–e) and the H&E staining (Figure 11b) are only an indication of the presence of cell membranes.

Whole blood erythrocyte and ghost-converted erythrocyte samples were analyzed by SEM-EDS, using backscattering electrons, showing 30 Pa at a voltage of 15 kV. Figure 13a,b shows an erythrocyte with an electro-dense structure, which was made evident by the dark color that can be observed in the internal part of the cell. This can be attributed to the Fe**^2+^** cationic content of the heme group of hemoglobin. Erythrocytes of approximately 9 µm in size are visible where the border of the cell membranes (light gray) contrasts with the internal part (dark gray) due to the constitution of the phospholipid bilayer of the cells. When the erythrocytes are lysed with the hypotonic medium and placed in an isotonic medium, self-sealing of the cell membrane occurs (Figure 13c,d), due to the amphipathic property of the membrane phospholipids. These self-seals will depend on the manipulation of the sample. In isolated erythrocyte ghosts, the membrane is evident without need of contrast to distinguish it from the red blood cell. Figure 13e presents an EDS analysis for normal red blood cells, the iron peak is visible at ≈1 and 6 KeV, and Figure 13f shows a ghost erythrocyte where iron is absent. Si in the samples remains unchanged due to the slide where the blood samples are mounted.

### 3.4. Biointeraction/Bioconjugation (AuNPs–Membrane Erythrocyte Ghost)

The SEM micrographs in Figure 14 show the interactions between monodispersed AuNPs (white dots) and ghost erythrocytes. The nanoparticles extend along the perimeter of the surface of the plasma membrane. EDS analysis (Figure 15) confirms the presence of the Au element in the cell membrane.

#### Biointeraction/Bioconjugation (AuNP-Fl-Membrane Erythrocyte Ghost)

The images of the ghost erythrocytes obtained by confocal microscopy show strong fluorescence due to the presence of AuNPs-FI (Figure 16). Through the fluorescent chemical compound, the AuNPs-FI-membrane erythrocyte ghost interaction is confirmed. Nanoparticles do not change the structure of erythrocyte ghosts. In the images, the symmetry of the cell membranes is highlighted by the fluorescence emitted.

## 4. Discussion

Synthesized AuNPs have a spherical shape with a normal and narrow particle size distribution and a face-centered cubic crystalline structure; the average diameter and crystalline structure obtained are typical for those obtained by the chemical reduction method, where a ruby red solution is obtained [42,43]. The solution color is indicative of the relationship between the SPR and AuNPs size in contact with visible light; depending on the area of the metallic nanoparticle, visible light diffracts within the electromagnetic spectrum range. Nanoparticles are made of pure Au. Electromagnetic excitation is called the polariton of the surface plasmon. For certain frequencies of the incident energy, there is a coupling between the frequencies of the incident wave and the oscillation of the polariton; this phenomenon is called surface plasmon resonance. Thus, for nanoparticles smaller than 30 nm, the reflected color is close to red, whereas in larger nanoparticles, the color shifts to purple because of the dispersion of shorter lengths when in colloidal suspension, as reported in the literature [36,44,45,46,47]. In the case of 13 nm AuNPs, the reddish solution presents a characteristic SPR peak at 525 nm. The negative surface charge is due to the presence of citrate on the nanoparticle surface, so this particle shape and size makes it possible to have a stable colloid where the particles are dispersed, representing the ideal condition for biomedical applications. Apparently, particle aggregation affects cytotoxic properties [37].

Fluorescein isothiocyanate adhered to the AuNPs, increasing the average particle size; however, colloid stability did not change, as FI increases the negative surface charge from −15.3 to −25.7 mV, achieving stabilization of the colloid. This interaction shifts the SPR absorption band of gold from l = 525 to l = 527 nm due to the electrophilic properties of the isothiocyanate group of fluorescein (N=C=S), as these are attracted by higher electron density areas. The negative charges of the free electrons in the metal conduction band and with the carboxyl groups (COO-) of the sodium citrate anion [48,49,50] expand along the length of the nanoparticle [51,52], where approximately 210 fluorophores are added for each nanoparticle. This addition is made possible by the Na^+^ group and electrostatic attraction of the oscillating AuNP electrons. This coupling of AuNPs-FI offers greater resolution for the detection of the interaction between nanoparticle and biological entity, using optical techniques [53].

When nanoparticles enter the human body, the first interaction is with red blood cells. This nanoparticle/biological entity interaction depends on colloidal forces, so the hemolysis test makes it possible to evaluate the rupture of the membranes of red blood cells due to the interaction of AuNPs. The greatest AuNP concentration (10,000 µg/mL) has a hemolytic degree close to 2%. This decreases in proportion to AuNP concentration, so that in terms of the ASTM F756-17 standard [38], this is considered to be a non-hemolytic material. This standard qualifies the hemolytic activity of materials into three types; non-hemolytic must have a percentage of less than 2%; slightly hemolytic materials must present a percentage of hemolysis between 2% and 5%; lastly, hemolytic materials must present a hemolysis percentage above 5%. This is because the phospholipid bilayer can adhere to AuNPs by binding charged nanoparticles so that they adhere to the heads of phosphocholine groups determined by a phosphate and choline electric dipole, P^−^-N^+^. Negatively charged nanoparticles (anionic) because of their functional group COO^-^ of sodium citrate interact preferentially with the positive end (N^+^); conversely, the positive charges (cationic) of the Na^+^ particle interact with the negative end of the dipole of the phosphocholine, stabilizing them on the surface of the erythrocyte membrane [54].

As hemoglobin belongs to a metalloprotein due to its Fe^2+^ group, it has three characteristic absorption peaks: α, β, and the main one, known as γ or Soret band [55] at l = 575, 540, and 412 nm, respectively, confirming the presence of the characteristic hemoglobin bands. During processing, washing eliminates hemoglobin, so that absorption decreases after each wash with lysis buffer from 100% to 8.71%, corroborating the loss of hemoglobin. A slight increase is noted after the penultimate wash, but this is due to the recurrences that remain prior to the addition of PBS solution. The color change in the solution from red to pink suggests that hemoglobin is released from the interior of the red blood cells, which when it clarifies is an indication of hemoglobin elimination or the presence of cell membranes (erythrocyte ghost), as previously reported [23,56,57,58]. As hemoglobin is eliminated during the lysis process, the Fe content decreases until finally disappearing when the process is completed.

The AuNPs–ghost erythrocyte interactions can be caused by electrostatic charges related to the carboxyl group (-COO) from sodium citrate onto the surface of the nanoparticle, with structures at the surface of the cell membrane [48]. Nanoparticles have great affinity with the cell membrane due to their size, which favors interaction with the membrane surface. Due to their antipathetic nature, they can become anchored to the surfaces, allowing the nanoparticle to pass through the cell without causing harm. This interaction can be mediated by endocytosis between particles 10 to 40 nm in diameter [59,60]. Due to the lack of specialized organelles in phantom erythrocytes, endocytic interactions cannot occur; thus, the interaction of the AuNP surface with the membrane is influenced by passive adhesion and accumulative interactions. 

When fluorophore is used, it adheres to the metallic nanoparticle and the surface of the plasma membranes in the phantom erythrocytes. The stability inherent in this combination relates to the attraction between charges of the nanoparticle electrons; the positive charges (Na^+^) and the negative charges (COO-) of the citrate, and the electrophilic charges of the isothiocyanate group (-N=C=S). This results in stability between the fluorophore and the metal. Similarly, fluorescein isothiocyanate causes dispersion, resulting in stability among the particles by creating a “shield” that prevents the electrons from moving out of the particle. By being adjacent to the membrane proteins, new protein ligands are produced: membrane proteins–fluorophore–particle–medium. This has already been reported but in the context of the electrostatic, hydrophobic, and hydrogen bonding interactions between BSA and functional groups (-NH_3_^+^, -CH_3_, -OH) [51]. In the case of AuNPs-FI, the (-N=C=S) group, similar to the (OH-) hydroxyl group of the molecule, is able to interact with membrane phospholipids. These polar heads are mostly constituents from specific types of charged groups: either (CH_3_CH_2_ N(CH_3_) _3_) phosphocholine, (HO-CH_2_-CH_2_-NH_2_) ethanolamine, or serine with its (OH-) terminal group. Binding-induced reorientation of the phosphocholine head group causes lipids in the fluid phase to have lower density in the P^−^ and N^+^ phosphocholine head group [54]. The fluorescent energy will be supplemented by the electronic excitation of the SPR, which induces a higher energy state [61,62] that is visible under a confocal microscope, confirming the AuNP-Fl-membrane erythrocyte ghost interaction. Nanobiointeraction comprises three dynamic components: (1) the surface of the nanoparticle that will be determined by the medium and functionalization at its surface, such as the shape, angle of curvature, porosity, crystallinity, heterogeneity, hydrophobicity, or hydrophilicity, (2) the interface (coating), and (3) the interaction between the environment and the biological zone [63,64]. The main effects resulting from the properties of the materials involve modification at the surface; these modifications interact with the environment of biological molecules and their cellular components. Thus, negatively charged Au nanoparticles interact with the cell membrane phospholipids by means of charge interaction. Due to the size of the AuNPs, these can be endocytosed as described above, but likewise, there may be a passive interaction with the depositions at the surface of the plasma membrane taking place in a way similar to that described by Guo et al. [65] and Hamadani et al. [66].

## 5. Conclusions

The inclusion of compatible fluorescein isothiocyanate to Au nanoparticles amplifies the spectral properties of gold nanoparticles, creating a stable and biocompatible colloid, which has the capacity to interact with biological entities.

Red blood cells can be transformed into phantom erythrocytes by removing hemoglobin, forming a cell membrane, which can be easily handled and is non-toxic to biological medium, meaning they can function as carrier vehicles.

AuNPs are deposited onto the surface of the ghost erythrocyte, where they interact with the phospholipids of the cell membrane. These interactions or bioconjunctions between hemoglobin-free erythrocytes, AuNPs, and FI indicate that these represent systems with the potential to encapsulate biological components that can be detected by the organic fluorophore, which is emitted as fluorescence in the presence of UV light. This offers a means to increase the sensitivity of monitoring, as well as improving sensing and release treatments, in the context of nanomedicine.

## Figures and Tables

**Figure 1 materials-14-06390-f001:**
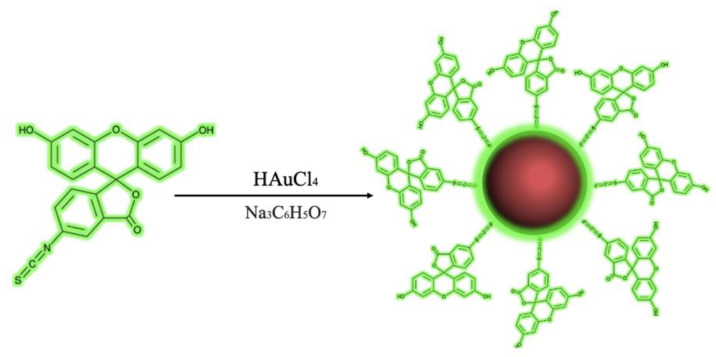
Representation of the addition of FI to the surface of the AuNPs in formation.

**Figure 2 materials-14-06390-f002:**
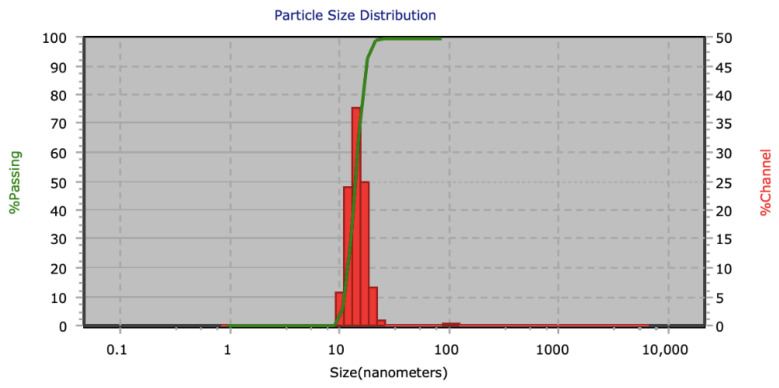
Size distribution of AuNPs obtained by DLS analysis.

**Figure 3 materials-14-06390-f003:**
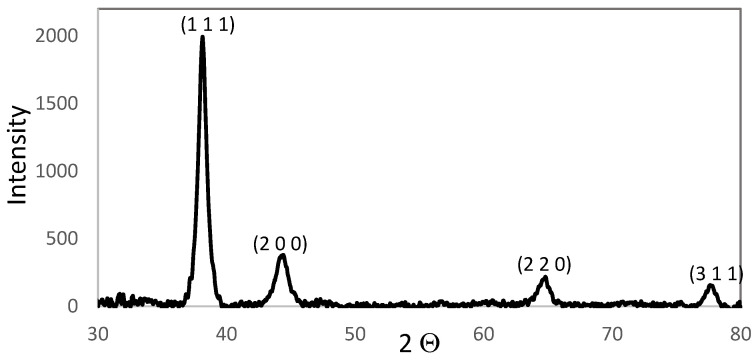
XRD pattern of AuNPs.

**Figure 4 materials-14-06390-f004:**
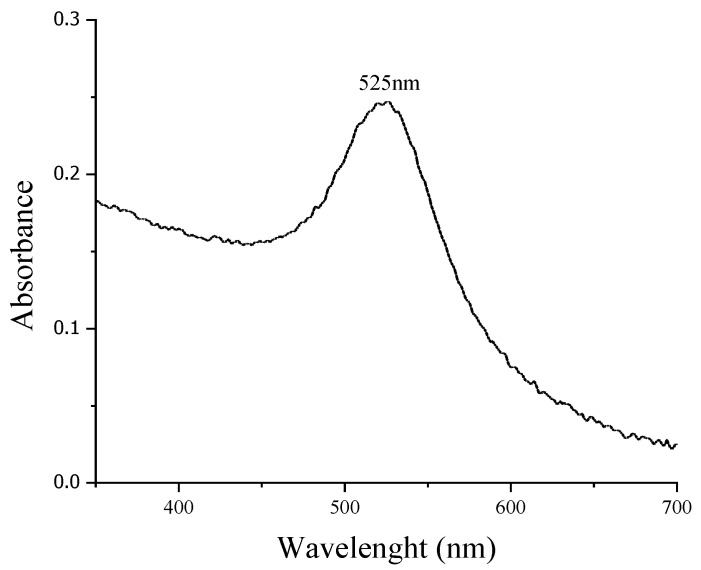
UV-Vis spectra of AuNPs colloid.

**Figure 5 materials-14-06390-f005:**
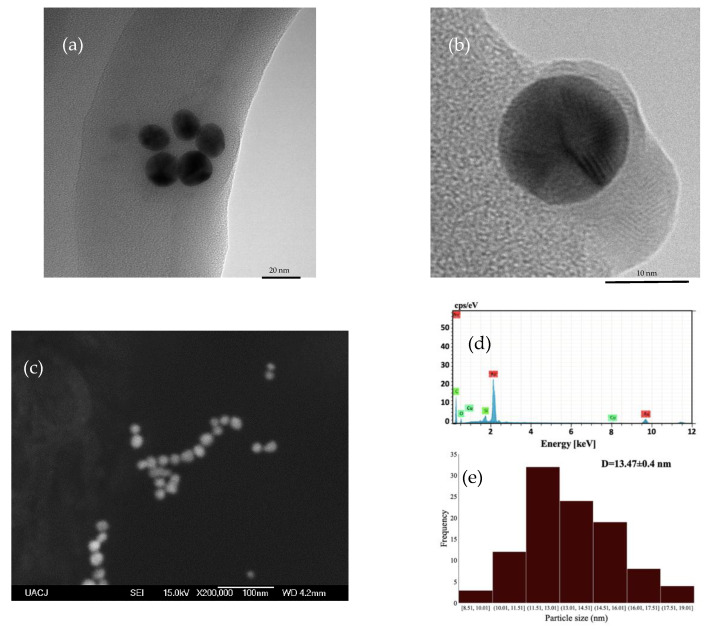
(**a**) TEM microscopy images of AuNPs, (**b**) a single Au particle, (**c**) SEM image of AuNPs, (**d**) EDS elemental analysis, and (**e**) particle size distribution of AuNPs.

**Figure 6 materials-14-06390-f006:**
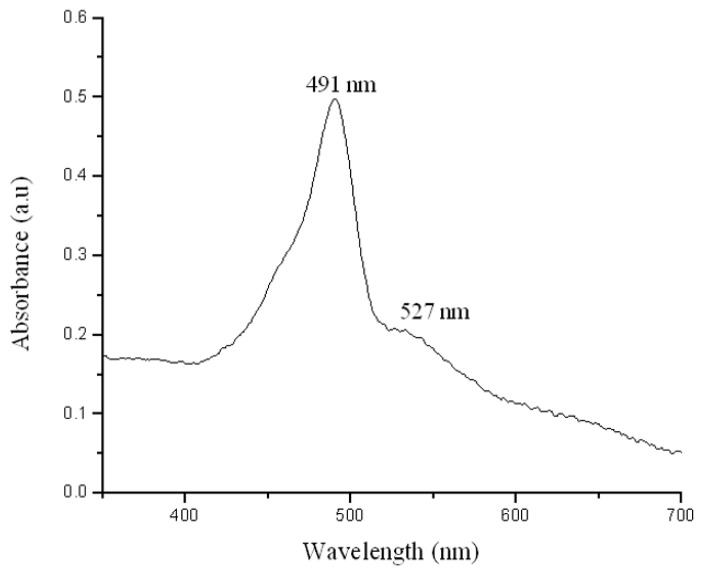
Absorbance plot for AuNPs-FI.

**Figure 7 materials-14-06390-f007:**
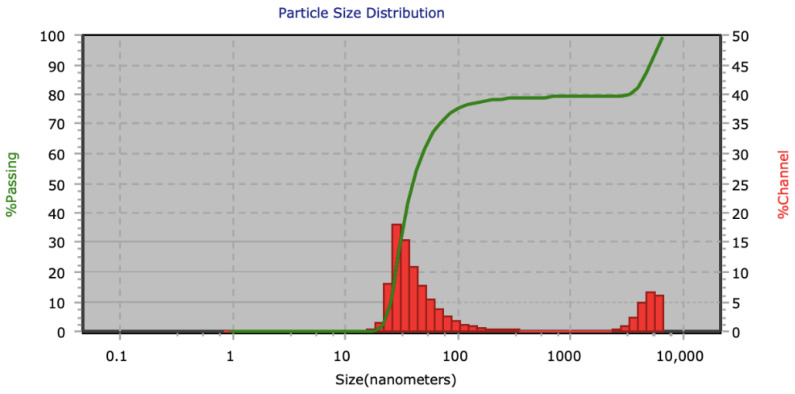
Particle size distribution of AuNPs-FI.

**Figure 8 materials-14-06390-f008:**
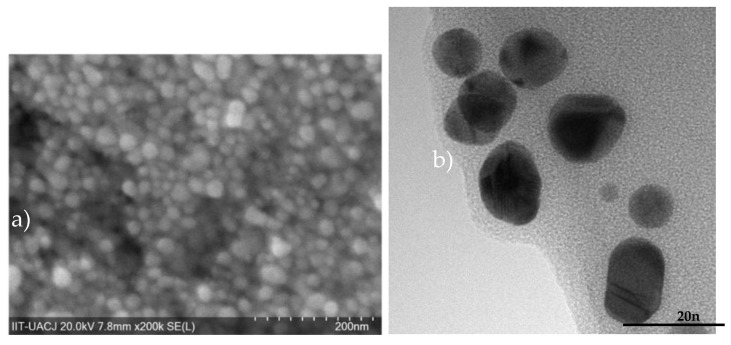
(**a**) SEM photo and (**b**) TEM image of AuNP-Fl.

**Figure 9 materials-14-06390-f009:**
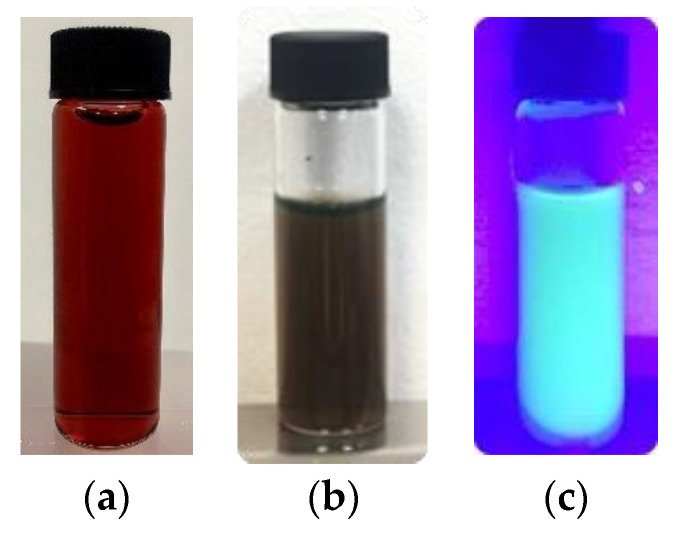
(**a**) Red ruby solution of AuNPs, (**b**) AuNPs-FI in visible light, and (**c**) AuNPs-FI exposed under ultraviolet light λ = 395 nm.

**Figure 10 materials-14-06390-f010:**
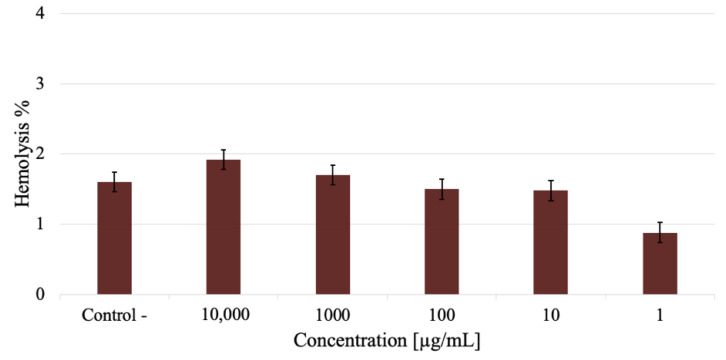
Hemolysis as a function of Au Nanoparticle concentration.

**Figure 11 materials-14-06390-f011:**
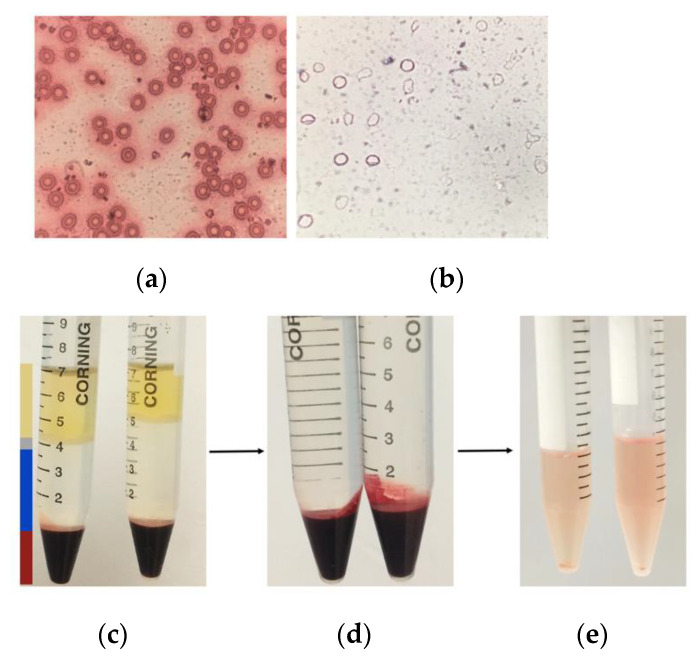
(**a**) Optical microscope images of red cells, (**b**) erythrocyte ghosts, (**c**) separation of blood components; yellow is serum, gray is lymphocytes, and red is erythrocytes. (**d**) Isolated erythrocytes and (**e**) erythrocyte ghost membranes.

**Figure 12 materials-14-06390-f012:**
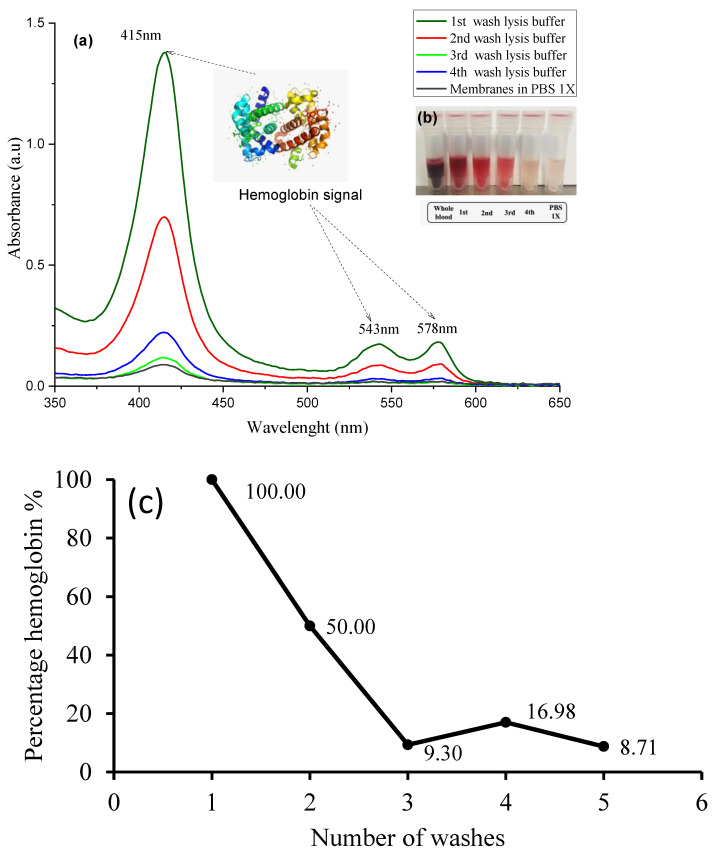
(**a**) UV-Vis absorption graph of the hemoglobin protein spectrum, (**b**) color variation and (**c**) hemoglobin concentration for each wash.

**Figure 13 materials-14-06390-f013:**
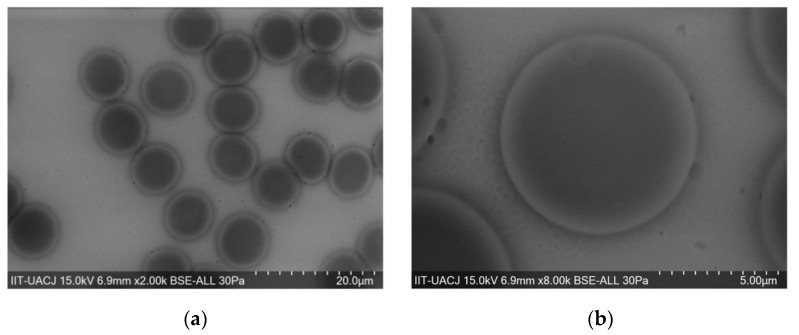
SEM micrographs of: (**a**,**b**) red blood cell, (**c**,**d**) ghost erythrocyte membranes, (**e**) EDS analysis of red blood cells and (**f**) EDS analysis of erythrocyte ghost.

**Figure 14 materials-14-06390-f014:**
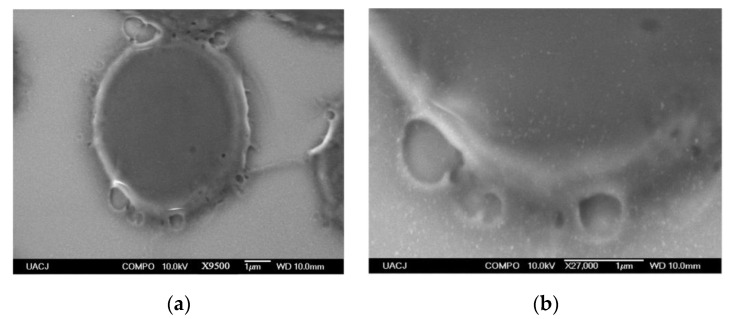
SEM micrographs of (**a**) isolated erythrocyte ghost and (**b**) membrane with AuNPs deposited on its surface.

**Figure 15 materials-14-06390-f015:**
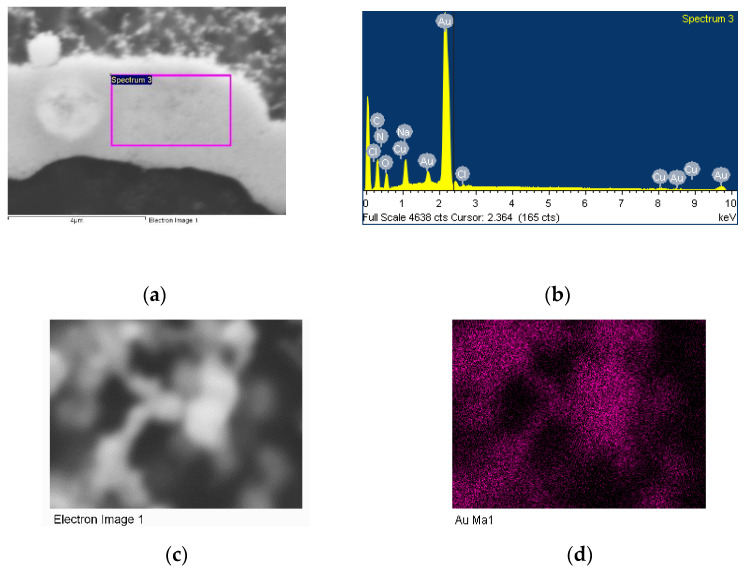
(**a**) SEM micrographs of erythrocyte membrane AuNPS-FI deposited on its surface, (**b**) EDS analysis onto cell membrane, (**c**) gold nanostructures attached to the erythrocyte, (**d**) EDS mapping of AuNPs.

**Figure 16 materials-14-06390-f016:**
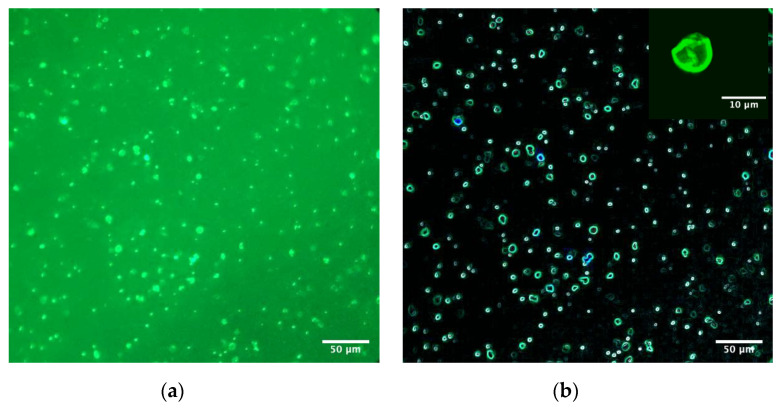
(**a**) Bright field and (**b**) dark field confocal image of erythrocytes ghosts with AuNPs-FI.

## Data Availability

The data presented in this study are available on request from the corresponding author.

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
