# Peer review of "Biointeraction of Erythrocyte Ghost Membranes with Gold Nanoparticles Fluorescents"

_materials, 2021, doi:10.3390/ma14216390_

Round 1
Reviewer 1 Report
The manuscript describes the preparation and characterization of the Au NPs modified with fluorescein (FITC) that can be binded to erythrocyte ghosts through electrostatic interaction of functional groups of sodium citrate COO- and fluorophore -N=C=S. The erythrocyte ghosts serve to “camouflage” AuNPs until they reach the target organ. This nanosystem is reported to be useful as a detection system for enhancing imaging sensitivity owing to SPR and fluorescence properties. The manuscript in its current form requires major revision. The following suggestions for improvement could be considered:
-The application of the proposed nanosystem is not explicitly described in the manuscript. Authors are encouraged to provide more details about expected application, as a detection system.
- Please carefully revise your manuscript to correct some typos. For example:
Line 77 reference notation style and same in Line 123, and in all of the manuscript.
Line 85 please change “ml” to “mL”, and in other parts of the manuscript.
Line 100-101 concentration notation and chemical formula.
Line 251, please correct iron ion notation.
etc.
-Please change the font style in Fig. 3, so it will be uniform for all figures. The quality of all figures should be revised and improved.
-In line 148, please provide an ICDD pattern number of face centered cubic planes of AuNPs.
-Authors are kindly requested to improve the resolution of SEM images. Is it possible for Authors to provide TEM images of AuNPs before and after coating with fluorescein?
-In the particle characterization section, please provide details of equipment used for SEI and EDS measurements.
-Authors are requested to use the same notation to the developed nanosystem either AuNPs-FI (line 207) or AuNPs-FITC (line 192). The provided different notations could be confusing to readers.
-In line 292 Authors refer to the ruby solution of AuNPs, so if possible they are requested to include the digital image of AuNPs solution, e.g. as inset in Fig. 6 or in Fig. 9.
-Please provide zeta potential and charge of AuNPs before and after conjugation to FITC, which can confirm the statement on colloid stability (Line 304).
- To what Authors referred in {Natalya Chekina] Line 313?
Authors are encouraged to revise figure captions and figures order as they are referred in the manuscript;:
-Please label particle size distribution in Fig. 5 as (d). Please add the label “EDS” to EDS spectra.
-Figure caption should refer to the data shown in the figure. Some figure captions are provided in the form of Result/Discussion, e.g. Fig. 10 caption.
-Fig. 12 caption should also provide a reference to the digital inset in Fig. 12a.
-Fig. 13 caption is very unclear from the perspective of magnification, e.g Fig,13b states that “red blood cells seen at 20 micrometer”, while the picture magnification is “5 micrometer”, etc. Further, EDS spectrum of Fig. 13e is referred to Fig. 13c. Please comment.
-Fig. 15 caption is incomplete, please revise.
-In the manuscript I haven’t found a sentence referring to citation [60] or does “60” (line 390) is a reference?
-Current conclusion more resembles discussion. Authors are encouraged to highlight their main results in the conclusion section.
Author Response
The application of the proposed nanosystem is not explicitly described in the manuscript. Authors are encouraged to provide more details about expected application, as a detection system.
To described the application the following text was added
In this work, we attached fluorescent gold nanoparticles onto the membranes of ghost erythrocytes in order to use them as drug carriers capable of detecting and sensing. These constructs can be used in Parkinson´s, a disease where significant erythrocyte extravasation has been diagnosed in the course of this illness, especially in the postcommissural putamen, thus these affected regions have been proposed as a possible therapeutic entrance for the pharmacological treatment of Parkinson´s disease [31]. Here, we propose that erythrocyte ghosts can act as carriers for drug treatment of any neurological disease like Parkinson’s that involves blood-brain barrier (BBB) fracture, for instance, in the treatment of accidents such as poisoning, where it is known that erythrocytes are extravasated in the postrema area [32] and in cerebrovascular accidents that involve brain hemorrhages. This construct has the capacity to interact with human tissue, showing great adaptability and affinity, reducing most or all the toxicity or damage that may result from the juxtaposition of the external material with the biological entity.
- Please carefully revise your manuscript to correct some typos. For example:
Line 77 reference notation style and same in Line 123, and in all of the manuscript.
The reference notation style was homogenized.
Line 85 please change “ml” to “mL”, and in other parts of the manuscript.
The notation was standardized at mL
Line 100-101 concentration notation and chemical formula.
Concentration notation and chemical formula was corrected
Line 251, please correct iron ion notation.
Iron notation was corrected
etc.
-Please change the font style in Fig. 3, so it will be uniform for all figures. The quality of all figures should be revised and improved.
All figures were uniformed and improved
-In line 148, please provide an ICDD pattern number of face centered cubic planes of AuNPs.
The JCPDS number was included
-Authors are kindly requested to improve the resolution of SEM images. Is it possible for Authors to provide TEM images of AuNPs before and after coating with fluorescein?
TEM images of AuNPs and AuNPs-FI were included, to obtain this additional analysis an extra co-author was included.
-In the particle characterization section, please provide details of equipment used for SEI and EDS measurements.
This text was included: (JSM-7000F, JEOL) from SEM and ( X-act EDS detector from Oxford) from EDS analysis.
-Authors are requested to use the same notation to the developed nanosystem either AuNPs-FI (line 207) or AuNPs-FITC (line 192). The provided different notations could be confusing to readers.
Al AuNPs-FX series was replaced with AuNPs-FI
-In line 292 Authors refer to the ruby solution of AuNPs, so if possible, they are requested to include the digital image of AuNPs solution, e.g. as inset in Fig. 6 or in Fig. 9.
The digital image of AuNPs solution was inserted in Fig. 9
-Please provide zeta potential and charge of AuNPs before and after conjugation to FITC, which can confirm the statement on colloid stability (Line 304).
Zeta potential measurement was included.
- To what Authors referred in {Natalya Chekina] Line 313?
It was a refence, [Natalya Chekina] was change for [47], it was added and renumbered the reference
Authors are encouraged to revise figure captions and figures order as they are referred in the manuscript;:
-Please label particle size distribution in Fig. 5 as (d). Please add the label “EDS” to EDS spectra.
The EDS spectra were labeled, the rest of the parts of the figure were labeled again.
-Figure caption should refer to the data shown in the figure. Some figure captions are provided in the form of Result/Discussion, e.g. Fig. 10 caption.
All figure captions were revised, the result and/or discussions were eliminated.
-Fig. 12 caption should also provide a reference to the digital inset in Fig. 12a.
It was provided
-Fig. 13 caption is very unclear from the perspective of magnification, e.g Fig,13b states that “red blood cells seen at 20 micrometer”, while the picture magnification is “5 micrometer”, etc. Further, EDS spectrum of Fig. 13e is referred to Fig. 13c. Please comment.
The SEM photos scale was remarked to clarify the erythrocyte size
-Fig. 15 caption is incomplete, please revise.
Figure caption was completed.
-In the manuscript I haven’t found a sentence referring to citation [60] or does “60” (line 390) is a reference?
That [60] reference should not be on line 390 so it was deleted
-Current conclusion more resembles discussion. Authors are encouraged to highlight their main results in the conclusion section.
The conclusions were modified.

Reviewer 2 Report
The paper by García Casillas et al describes the biointeraction of erythrocyte ghost membranes with fluorescently-labeled nanoparticles. The paper is clear and in my opinion deserves publication.
In the introduction i would mention to different examples of techniques frequently used for the implementation of nanosensors so as carbon nanotube (chem rev 2019, 119, 599-663) or Metal-organic frameworks (adv. mater. 2018, 30, 1704303) or sol gel methods (ACS omega 2018, 3, 17319; Inorganica chimica acta 2008, 361, 1116-1121).
Author Response
In the introduction i would mention to different examples of techniques frequently used for the implementation of nanosensors so as carbon nanotube (chem rev 2019, 119, 599-663) or Metal-organic frameworks (adv. mater. 2018, 30, 1704303) or sol gel methods (ACS omega 2018, 3, 17319; Inorganica chimica acta 2008, 361, 1116-1121).
the text and citations were added
Reviewer 3 Report
1) Line 44: I suggest changing the phrase “intermediates such as fluorescein” to “fluorophores”; unless there is something special about fluorescein and/or intermediates other than fluorophores can enhance the effect.
2) Lines 45-55: The flow of text is quite incoherent. This part of introduction is not developed in a logical manner, but presented in somewhat of a random manner that includes important mechanistic information that are not presented adequately/correctly. Specifically, some of the mechanisms related to metal-enhanced fluorescence is presented superficially, and to some extent, incorrectly. Authors may wish to examine some of the previous literature in this area. In particular, some of these previous publications must be cited: K. Aslan et al, Curr. Opin. Biotech., 15, 55-52, 2005;
3) There are several papers describing the use of RBCs as platform for the delivery of various agents, including their use in photo-activation. These papers must also be cited: L. Koleva et al., Pharmaceutics, 12, 276, 2020., T. Hanley et al, Biomolecules, 11, 729, 2021.
4) From the Introduction, I really do not understand what the main objective of this study is and what it adds to the field. This needs to be much better indicated.
5) Figures 4, 6, 12: Absorbance cannot have arbitrary units. It is a unitless quantity where the values actually have physical meanings.
6) EM images are not of high quality; they are quite blurry.
7) According to Figure 12, the ghosts clearly contain some residual hemoglobin. The Q and Soret bands are evident. Authors need to acknowledge that it is very difficult to completely deplete the hemoglobin from human RBCs.
8) The study is focused on characterization of these particles. But what is really the point of this study? There is no real discussion of the significance.
Author Response
1) Line 44: I suggest changing the phrase “intermediates such as fluorescein” to “fluorophores”; unless there is something special about fluorescein and/or intermediates other than fluorophores can enhance the effect.
This sentence was changed.
2) Lines 45-55: The flow of text is quite incoherent. This part of introduction is not developed in a logical manner, but presented in somewhat of a random manner that includes important mechanistic information that are not presented adequately/correctly. Specifically, some of the mechanisms related to metal-enhanced fluorescence is presented superficially, and to some extent, incorrectly. Authors may wish to examine some of the previous literature in this area. In particular, some of these previous publications must be cited: K. Aslan et al, Curr. Opin. Biotech., 15, 55-52, 2005;
The text was improved, and corrected the mechanisms related to metal-enhanced. The literature was examined.
3) There are several papers describing the use of RBCs as platform for the delivery of various agents, including their use in photo-activation. These papers must also be cited: L. Koleva et al., Pharmaceutics, 12, 276, 2020., T. Hanley et al, Biomolecules, 11, 729, 2021.
The paper was cited.
4) From the Introduction, I really do not understand what the main objective of this study is and what it adds to the field. This needs to be much better indicated.
The introduction was modify and the main objective was improved.
5) Figures 4, 6, 12: Absorbance cannot have arbitrary units. It is a unitless quantity where the values actually have physical meanings.
The units were eliminated.
6) EM images are not of high quality; they are quite blurry.
7) According to Figure 12, the ghosts clearly contain some residual hemoglobin. The Q and Soret bands are evident. Authors need to acknowledge that it is very difficult to completely deplete the hemoglobin from human RBCs.
It is true, this fact was accepted.
8) The study is focused on characterization of these particles. But what is really the point of this study? There is no real discussion of the significance.
The application of the AuNPs-FI-erythrocyte system was included as well as the advantages of the compound in this text
In this work, we attached fluorescent gold nanoparticles onto the membranes of ghost erythrocytes in order to use them as drug carriers capable of detecting and sensing. These constructs can be used in Parkinson´s, a disease where significant erythrocyte extravasation has been diagnosed in the course of this illness, especially in the postcommissural putamen, thus these affected regions have been proposed as a possible therapeutic entrance for the pharmacological treatment of Parkinson´s disease [33]. Here, we propose that erythrocyte ghosts can act as carriers for drug treatment of any neurological disease like Parkinson’s that involves blood-brain barrier (BBB) fracture, for instance, in the treatment of accidents such as poisoning, where it is known that erythrocytes are extravasated in the postrema area [34] and in cerebrovascular accidents that involve brain hemorrhages. This construct has the capacity to interact with human tissue, showing great adaptability and affinity, reducing most or all the toxicity or damage that may result from the juxtaposition of the external material with the biological entity.

Round 2
Reviewer 1 Report
The manuscript has become well-organized.
Author Response
The manuscript has become well-organized.
thank you. Best regards
Reviewer 3 Report
Manuscript is somewhat improved. My concern remains about lack of a solid discussion on the significance and the utility of these particles. Authors have indicated the use of the particles as drug carriers (e.g., for treatment of some neurological disorders). I don't see anything in this manuscript that relates to such applications. RBCs are just labeled with FITC-conjugated gold nanoparticles. Not sure how these particular constructs provide the intended therapeutic capability as the therapeutic agent/component is not identified. The gold nanoparticles? FITCS? Not sure if any of these two in themselves have therapeutic potential. There is no therapeutic agent in these constructs as they are.
Author Response
There is no therapeutic agent in these constructs as they are.
Thank you for the observation, we accept and agree with this comment, nonetheless the title of the paper is “Biointeraction of erythrocyte ghost membrane with AuNPs- Fluorescents”, our goal was to develop an autologous construct and that is what we are presenting, a potential vector that needs to be tested in future experiments, still, we think that this construct/vector by itself deserves to be known by the scientific community with the hope that other researchers and laboratories might test this construct with their own drugs, meanwhile, we are waiting to overpass the pandemic restriction to test this construct in vivo experiments with a panel of therapeutic drugs. We think that the idea of the therapeutic potential of ghost-AuNPs is important in the discussion to encourage others to test this vector.
Furthermore, gold nanoparticle functions as a therapeutic agent, as mentioned by Nair et al. 2015; Zhai et al. 2017; Zhang et al. 2018; Luan et al. 2020, where optical properties of gold particles were used as a therapeutic agent in tumor cells. Govindaraju et al. (2017) used the particles for brain sensing purposes.